# Risk Factors and Outcomes Following Septostomy during Fetoscopic Surgery for Twin-to-Twin Transfusion Syndrome

**DOI:** 10.3390/jcm10163693

**Published:** 2021-08-20

**Authors:** Clifton O. Brock, Eric P. Bergh, Edgar A. Hernandez-Andrade, Rodrigo Ruano, Anthony Johnson, Ramesha Papanna

**Affiliations:** Department of Obstetrics, Gynecology, and Reproductive Sciences, McGovern Medical School, University of Texas Health Science Center at Houston, 6431 Fannin Street, Houston, TX 77030, USA; clifton.o.brock@uth.tmc.edu (C.O.B.); eric.p.bergh@uth.tmc.edu (E.P.B.); edgar.a.hernandezandrade@uth.tmc.edu (E.A.H.-A.); rodrigo.ruano@uth.tmc.edu (R.R.); anthony.johnson@uth.tmc.edu (A.J.)

**Keywords:** prenatal ultrasound, fetal medicine, fetal surgery, placental laser surgery, monochorionic diamniotic twins, twin-to-twin transfusion syndrome, septostomy

## Abstract

Fetoscopic laser photocoagulation (FLP) of placental anastomoses is the preferred treatment for twin-to-twin transfusion syndrome (TTTS). Iatrogenic septostomy (IOS) during FLP is associated with increased risk of neonatal morbidity and mortality. We sought to identify risk factors for IOS and quantify the resultant outcomes. This is a secondary analysis of prospectively collected cases of TTTS in monochorionic diamniotic twins following FLP at a single center. Pre-operative ultrasound characteristics and operative technique (i.e., cannula size, total energy used) were compared between cases with vs. without IOS. Pregnancy and neonatal outcomes were also compared. Of 475 patients that had FLP, 33 (7%) were complicated by IOS. There was no association between operative technique and IOS. IOS was more common with later diagnosis, but less likely when selective fetal growth restriction (sFGR) was present. Survival was similar between groups (76% vs. 76% dual survivors, *p* = 0.95); however, IOS was associated with earlier delivery (29.7 vs. 32.0 wks, *p* < 0.01) and greater composite neonatal morbidity (25% vs. 8% in both twins, *p* = 0.02). Risks of IOS at greater gestational ages without sFGR may be related to a larger collapsed intervening membrane area and the resulting increased risk of puncture on entry.

## 1. Introduction

Twin-to-twin transfusion syndrome (TTTS) is a complication affecting 10–15% of monochorionic diamniotic (MCDA) twin pregnancies [1,2,3]. Fetoscopic laser photocoagulation (FLP) of the anastomotic placental blood vessels between twins has become the standard of care for management of TTTS. FLP is associated with increased survival over previously recommended modalities including serial amnioinfusion and purposeful septostomy [4,5]. Despite this, FLP is associated with several complications related to the intervening and surrounding fetal membranes including preterm premature rupture of membranes (PPROM), chorion amnion separation (CAS), and inadvertent iatrogenic septostomy (IOS) [6,7].

IOS specifically, defined as perforation of the intervening membranes, has more recently been regarded as a complication of FLP rather than a possible treatment (Figure 1). The reported incidence of IOS is 1.3–20% [6,8,9,10,11]. It has been associated with increased rates of preterm delivery and PPROM at earlier gestational ages, development of an iatrogenic pseudo-monoamniotic pregnancy with associated cord entanglement, amniotic band syndrome, decreased survival, and increased neonatal morbidity [8,9,10,11]. IOS is thought to occur either by heat degeneration during ablation of anastomoses located behind the intervening membrane (i.e., the anastomoses are on the donor side of the membrane), direct inadvertent puncture during FLP (i.e., during attempts to manipulate donor position), or puncture of the collapsed donor membrane reflection upon entry into the recipient sac (i.e., traversing the donor sac). Currently, few data exist to elucidate the risk factors associated with IOS, and only a small number of studies have reported on perinatal morbidity following this complication [9,11].

In the present study, we sought to determine whether risk factors for IOS, related to demographics or ultrasound findings upon diagnosis of TTTS, could be identified. We also examined the specific operative techniques employed with the same goal. Finally, we compared delivery and neonatal outcomes between those with FLP complicated by, versus not complicated by IOS.

## 2. Materials and Methods

This is a secondary analysis of an ongoing prospective cohort study of patients that underwent FLP for TTTS at the University of Texas Health Science Center in Houston, TX (UTHealth), McGovern Medical School. This study was approved by the UTHealth Institutional Review Board (IRB#: HSC-MS-19-0189). Patients referred to the Fetal Center at UTHealth and confirmed to have a diagnosis of TTTS who then underwent FLP between September 2011 and July 2020 were included. Cases with higher-order multiples as well as those that had selective reduction at the time of attempted FLP were excluded. All patients were consented for the study prior to FLP.

Patients who presented to our center for suspected TTTS underwent a comprehensive anatomy ultrasound and evaluation for TTTS. Estimated fetal weights (EFWs) and Doppler velocimetry studies of the umbilical artery and vein, middle cerebral artery, and ductus venosus waveforms were assessed, as were the bladders of each twin. TTTS was diagnosed in the setting of concomitant polyhydramnios (maximum vertical pocket (MVP) > 8 cm) of the recipient twin and oligohydramnios (MVP < 2 cm) of the donor twin. The stage of TTTS was determined as described by Quintero et al. [12]. FLP was offered to all patients that had Stage II–IV TTTS. FLP was offered for Stage I TTTS when either a short cervix (<1.5 cm early in the study period and <3.0 cm over the last 3 years) or severe symptoms of polyhydramnios were present. Selective fetal growth restriction (sFGR) was diagnosed when growth discordance between twins greater than 25% was observed, and twin anemia polycythemia sequence (TAPS) was diagnosed when the middle cerebral artery peak systolic velocity was greater than 1.5 multiples of the median (MoM) in the anemic twin and less than 1.0 MoM in the polycythemic twin [13].

FLP was carried out under local anesthesia with sedation unless a trocar was used for anterior placentation that required laparoscopic assistance, in which case general anesthesia was administered. A 9, 10, 11, or 12-French cannula was placed into the recipient sac under direct ultrasound visualization using either a metal trocar for entry or a guidewire introduced through an 18-gauge needle (i.e., Seldinger technique). Using either a Karl Storz 0° or 30°degree rod lens operative fetoscope with a diode laser (400- or 600-micron fiber), selective photocoagulation of anastomotic vessels was carried out followed by application of the Solomon technique when possible [14]. Selective reduction was offered prior to surgery in case there was failure to complete FLP or an agonal fetal heart rate occurred intraoperatively. (Selective reduction was not offered in the former situation for gestational ages greater than or equal to 22 weeks.) All patients were hospitalized the night following FLP.

IOS was diagnosed either intraoperatively or on post-operative ultrasound (post-operative day one). The diagnosis was assigned when one or more of the following observations were made: (i) a free-floating membrane with an area of visible disruption, (ii) equalization of the volume of amniotic fluid on donor and recipient sides of the membrane (i.e., similar MVPs) combined with equalization of the fluid echogenicity on each side of the membrane, (iii) fetal parts of the donor or recipient twin (including segments of umbilical cord) crossing into the opposite twin’s sac. Patients were discharged from the hospital the morning after surgery following post-operative ultrasound. Serial ultrasounds were recommended to be carried out by the referring physician weekly following FLP for six weeks including assessment of the amniotic fluid, membranes, and Doppler velocimetry studies. After the initial six weeks, routine biweekly ultrasound surveillance was recommended. Data on demographics, ultrasound findings at initial consultation and the day after surgery, intraoperative findings and technical aspects of FLP were collected prospectively prior to discharge. Data on any subsequent antepartum hospitalizations, delivery outcomes, and neonatal outcomes were collected from the referring physician as well as the hospital where delivery occurred.

Patients with vs. without IOS were compared. Demographic and initial ultrasound characteristics, including age, body mass index (BMI), parity, gestational age at fetal intervention, Quintero stage, placental location, donor/recipient MVPs, cervical length, and concomitant TAPS/sFGR, were compared to determine possible risk factors for IOS. Surgical characteristics including the location and type of uterine entry, cannula size, total energy used, number of anastomoses ablated, total laser time, application of the Solomon technique, and amnioinfusion/amnioreduction were similarly compared between groups to determine whether the choice of operative technique was associated with IOS. Pregnancy outcomes including chorion amnion separation (CAS), preterm premature rupture of membranes (PPROM), gestational age at delivery, single/dual demise, indication for delivery and mode of delivery were also compared. For neonatal outcomes, a composite morbidity index including bronchopulmonary dysplasia (BPD), Stage III/IV intraventricular hemorrhage (IVH), culture-proven sepsis, and necrotizing enterocolitis (NEC) was constructed. Transient tachypnea of the newborn (TTN), respiratory distress syndrome (RDS), and neonatal death were analyzed separately. To prevent incorrect use of statistical tests due to non-independence of observations, donor and recipient outcomes were analyzed separately.

Standard univariate statistics were performed to compare demographic, diagnostic, and surgical data between groups. For continuous variables, the Student’s *t*-test or the Wilcoxon rank sum test was used depending on whether the variable approximated a normal distribution. For discrete variables, chi-squared tests or Fisher exact tests were used depending on the expected values in the relevant contingency table.

## 3. Results

There were 475 patients meeting inclusion criteria that underwent FLP for TTTS during the study period. Among these, 33 (6.9%) had an IOS, while the remaining 442 (93.1%) did not. Patients in each group had similar age, BMI, and parity (Table 1). There were also no differences in the Quintero stage at diagnosis or the placental location between groups. However, patients that had IOS underwent FLP at later gestational ages (21.9 vs. 20.3 weeks, *p* < 0.01), with greater recipient MVPs and greater EFWs for both twins. Patients that had IOS were also less likely to have a fetus with sFGR (24.2% vs. 45.2%, *p* = 0.02). Of note, the cervical length was slightly less in patients that had IOS (33.5 vs. 38.0 cm, *p* = 0.03).

Details regarding the choice of operative technique for FLP are shown in Table 2. The technique of entry into the recipient’s amniotic sac (Seldinger vs. metal trocar) and the entry site on the maternal abdomen were not associated with IOS. Cases where two uterine punctures were required were more commonly associated with IOS; however, the difference was not statistically significant (3.0% vs. 1.6%, *p* = 0.53). There was no difference in the total number of anastomoses ablated, total amount of energy used, total laser time, or total time with the cannula in the recipient twin’s amniotic sac between the two groups. Inability to completely Solomonize the placenta was more common in cases with IOS, but this also was not a statistically significant finding (6.1% vs. 1.4%, *p* = 0.07).

Following FLP, overall survival rates were similar between groups (Table 3). Rates of CAS and PPROM were also similar between groups; however, PPROM prior to 28 weeks was more common following IOS (39.4% vs. 21.3%, *p* = 0.02). The interval between FLP and PPROM and the length of latency following PPROM were similar between groups (Figure 2C,D). The gestational age of delivery following iatrogenic IOS was 2.3 weeks earlier than in patients that did not have IOS (Figure 2A, 29.7 vs. 32.0 weeks, *p* < 0.01). Patients with IOS had a shorter FLP-to-delivery interval (Figure 2B, 8.1 vs. 11.1 weeks, *p* < 0.01) and were more likely to be delivered for preterm labor (48.5% vs. 34.2%, *p* < 0.01). Iatrogenic delivery for fetal indication, including pseudo-monoamniotic twins, was similar between groups (12.1% vs. 12.6%). There were no differences in the mode of delivery, and the two groups had similar rates of antepartum donor and recipient demise as well as dual demise. The risk of iatrogenic monoamnionicity following IOS was 21.1%, while apparent spontaneous monoamnionicity occurred in 0.5% of cases without IOS based on follow-up ultrasounds performed by the referring physician.

While pregnancy outcome data and neonatal survival data were available for all 475 patients, data on neonatal morbidity could not be obtained for 28 (5.9%) pregnancies (56 neonates). Therefore, of 950 potential neonates, 803 were available for analysis after accounting for intrauterine demises, termination of pregnancy, and loss to follow-up, as detailed in Figure 3. While there was no difference in overall survival following IOS (Table 3), the likelihood of composite morbidity in survivors was greater after IOS (Table 4, morbidity in both twins in 25.1% vs. 8.1%, *p* = 0.02). Rates of BPD were greater among donors (33.3% vs. 10.0%, *p* < 0.01), with a similar, but statistically insignificant, trend among recipients (17.2% vs. 7.2%, *p* = 0.07). Rates of neonatal death were similar between groups. There were no differences in any other individual neonatal outcomes between groups.

## 4. Discussion

The present study includes several important findings: (1) IOS occurred in 6.9% of TTTS cases treated with FLP. (2) IOS was more likely to occur in TTTS treated at later gestational ages with greater recipient MVPs, larger EFWs, and greater growth discordance between twins. Hence, IOS was less common in the setting of sFGR. (3) No choice of operative technique, including entry method or site, amount of energy used, operative time, cannula size, or application of the Solomon technique, was associated with IOS. (4) There was no change in overall survival in pregnancies complicated by IOS. (5) Pregnancies with IOS were delivered at earlier gestational ages and were associated with greater composite perinatal morbidity, largely due to BPD.

The prevalence of IOS after FLP in this study (6.9%) was similar to that found in studies conducted at other large, experienced centers that perform the operation [8,10,11]. An exception is the IOS rate reported by Peeters et al. (20%), where nearly half of IOS was observed between 5 and 68 days after FLP [9]. Other studies only reported IOS observed 24–72 h after FLP [8,10]. The correct criteria to diagnose IOS as time progresses from FLP are less clear as fluid is expected to equalize within weeks following successful FLP, and spontaneous disruption of the membrane has been described in otherwise uncomplicated MCDA pregnancies [15,16]. We did not observe a difference in overall survival, which was decreased following IOS in two prior studies [8,11]. We also did not observe higher rates of fetal demise following IOS as reported by Cruz-Martinez et al. and Li et al. [10,11]. The reasons for these differences are not currently clear.

This study is unique in that data from each operative step of FLP were evaluated. Our results suggest that the size of the cannula used for entry into the recipient sac, total energy used, total cannula time, and technique and site of entry do not appear to contribute to the likelihood of IOS. As with other studies, we found that IOS is more likely to occur in cases where FLP is performed at later gestational ages [9,10,11]. However, we also found that sFGR and a lower recipient MVP may be associated with a decreased likelihood of IOS. These observations may imply a lower risk of IOS in cases where the collapsed donor membrane takes up less area in the possible uterine window for entry into the recipient’s gestational sac. While IOS due to perforation of the collapsed intervening membrane has been described, we are not aware of prior data linking growth discordance between twins to an increased risk of IOS. Care in planning the entry site far from the donor, particularly at later gestational ages, may help avoid IOS. It is not clear what proportion of IOS is related to this complication vs. other mechanisms (i.e., ablation through the dividing membrane, direct puncture with the laser fiber or camera). While a slightly shorter cervical lengths were observed in patients that had IOS, this is likely related to the greater gestational age at which they underwent FLP.

Data on neonatal morbidity following IOS are sparce. Peeters et al. reported increased rates of severe cerebral injury, whereas Li et al. reported an increased likelihood of neonatal cerebral imaging anomalies following IOS [9,11]. The present study had similar rates of IVH regardless of IOS; however, the additional findings on postnatal brain imaging (i.e., intraventricular echodensities, parenchymal cysts) reported by Li et al. and Peeters et al. were not consistently available for us to analyze. We found that composite morbidity is greater following IOS as are rates of BPD specifically, particularly in the ex-donor twin. These differences are likely related to earlier gestational ages of delivery following IOS. Well-designed prospective studies may help better quantify the burden of neonatal complications associated with IOS. There are also few data on the indications for delivery following IOS. Increased rates of PPROM prior to 28 weeks suggest that membrane complications may underlie resultant preterm labor, whereas rates of delivery related to iatrogenic monoamnionicity seem to play a smaller role. The specific mechanism by which septostomy may lead to preterm labor is currently unclear. It is also difficult to ascertain whether IOS is truly the cause of resultant complications or a surrogate for other complicating factors, such as technically difficult surgeries requiring greater manipulation. This may be suggested by the higher (but not statistically significant) rates of a second entry and inability to complete Solomonization in the IOS group. Well-designed prospective studies may help clarify this question.

Strengths of the present study include the large number of patients from a single center with extensive experience in FLP. This is also the first study of which we are aware where identification of risk factors for IOS, including analysis of extensive intraoperative variables, was performed. This study is not without limitations. While demographic, intraoperative, and initial postoperative ultrasound data were prospectively collected, follow-up data on neonatal outcomes were collected from the referring physicians and delivery hospitals, which have heterogeneous electronic medical record systems and workflows for recording clinical data. Furthermore, the specific mechanism of IOS cannot always be determined with certainty as data on whether the laser was fired through the intervening membrane were not routinely collected and septostomy was often discovered on postoperative ultrasound.

## 5. Conclusions

In summary, IOS is associated with increased risk of delivery at earlier gestational ages and composite neonatal morbidity, but not overall survival in the current population. Larger areas of the collapsed intervening membrane may be a risk factor IOS. Further studies with prospective measurements of distances between the recipient sac entry site and the donor twin are required to further evaluate this hypothesis. Further work will also be required to understand the mechanism linking IOS to spontaneous preterm labor. Meanwhile, it may be possible to reduce rates of IOS by careful planning of uterine entry with specific consideration of the donor’s location. Purposeful septostomy as part of FLP is likely best avoided.

## Figures and Tables

**Figure 1 jcm-10-03693-f001:**
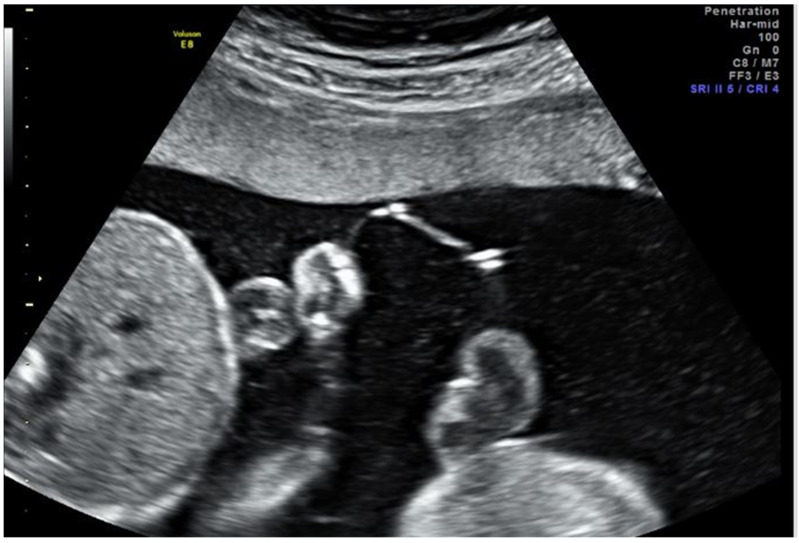
Ultrasound image the day after fetoscopic laser photocoagulation (FLP) showing the intervening membranes with a small defect, fluid equilibration between sacs, and similar fluid echogenicity between sacs.

**Figure 2 jcm-10-03693-f002:**
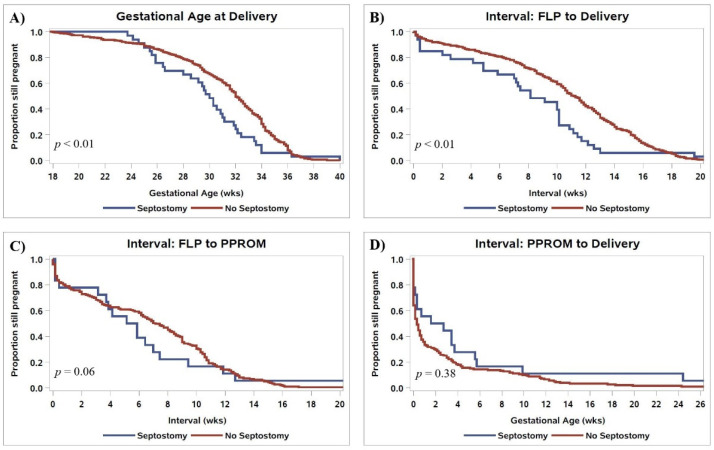
Kaplan–Meier curves comparing (**A**) gestational age of delivery, (**B**) the interval between FLP and delivery, (**C**) the interval from FLP to PPROM, and (**D**) the interval from PPROM to delivery in patients who had vs. did not have IOS. *p*-values are for log-rank tests comparing survival distributions.

**Figure 3 jcm-10-03693-f003:**
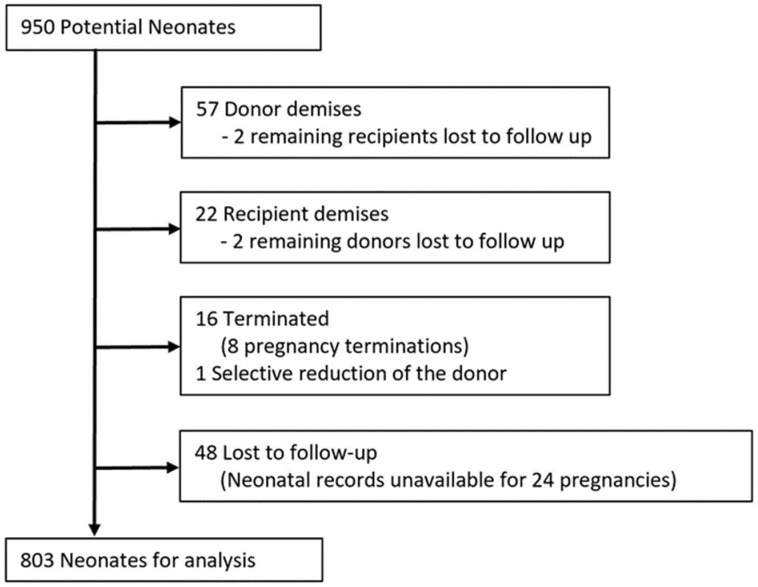
Flow chart showing neonates available for analysis.

**Table 1 jcm-10-03693-t001:** Demographic and initial ultrasound characteristics.

Characteristic	No Septostomy (*n* = 442)	Septostomy (*n* = 33)	*p*-Value
Maternal Age	28.5 ± 5.5	28.2 ± 5.3	0.70
BMI	27.1 (23.4, 31.9)	26.0 (22.1, 30.0)	0.27
Nulliparous	185 (41.9%)	15 (45.5%)	0.69
Gestational Age at Intervention	20.3 (18.6, 22.1)	21.9 (20.4, 23.7)	<0.01
Quintero Stage of TTTS			
I	54 (12.2%)	5 (15.2%)	0.81 *
II	127 (28.7%)	9 (27.3%)	
III	243 (55.0%)	17 (51.5%)	
IV	18 (4.1%)	2 (6.1%)	
Placental Location			
Anterior	238 (53.8%)	16 (48.5%)	0.66
Posterior	189 (42.8%)	15 (45.5%)	
Other	15 (3.4%)	2 (6.1%)	
Recipient MVP (cm)	10.7 (9.0, 13.3)	12.5 (10.6, 14.0)	<0.01
Donor MVP (cm)	0.9 (0.0, 1.7)	1.0 (0.0, 1.9)	0.77
Recipient EFW (g)	329 (239, 456)	446 (317, 599)	<0.01
Donor EFW (g)	245 (181, 361)	366 (244, 454)	<0.01
TAPS	68 (15.4%)	3 (9.1%)	0.45 *
sFGR	200 (45.2%)	8 (24.2%)	0.02
CL (mm)	38.0 (30.0, 45.0)	33.5 (28.0, 40.0)	0.03

* Fisher’s exact test used due to small expected values in the contingency table. Data expressed as: *n* (%), mean ± standard deviation, or median [1st quartile, 3rd quartile]. Abbreviations: BMI—body mass index, CL—cervical length, EFW—estimated fetal weight, MVP—maximum vertical pocket, sFGR—selective fetal growth restriction, TAPS—twin anemia polycythemia sequence, TTTS—twin-to-twin transfusion syndrome.

**Table 2 jcm-10-03693-t002:** Operative characteristics of fetoscopic laser photocoagulation.

Operative Characteristic	No Septostomy (*n* = 442)	Septostomy (*n* = 33)	*p*-Value
Type of cannula entry			
Seldinger	276 (62.4%)	23 (69.7%)	0.41
Metal trocar	166 (37.6%)	10 (30.3%)	
Number of punctures			
1	435 (98.4%)	32 (97.0%)	0.44 *
2	7 (1.6%)	1 (3.0%)	
Entry site			
RUQ	110 (25.3%)	12 (36.4%)	0.50 *
RLQ	55 (12.7%)	3 (9.1%)	
LUQ	103 (23.7%)	4 (12.1%)	
LLQ	88 (20.3%)	6 (18.2%)	
Upper midline	20 (4.6%)	1 (3.0%)	
Lower midline	42 (9.7%)	6 (18.2%)	
Laparoscopic	7 (1.6%)	0 (0.0%)	
Other/unknown	9 (2.1%)	1 (3.0%)	
Cannula size			
9 Fr	37 (8.4%)	7 (21.2%)	0.10 *
10 Fr	144 (32.6%)	6 (18.2%)	
11 Fr	10 (2.3%)	0 (0.0%)	
12 Fr	248 (56.1%)	20 (60.6%)	
Unknown	3 (0.7%)	0 (0.0%)	
Number of anastomoses ablated	10 (8, 13)	11 (6, 12)	0.52
Total energy (kJ)	4.0 (3.0, 5.2)	4.4 (2.8, 5.9)	0.48
Laser time (min)	2.6 (2.1, 3.5)	3.2 (2.2, 4.1)	0.20
Solominization			
Performed	420 (95.0%)	30 (90.9%)	0.15 *
Not performed	16 (3.6%)	1 (3.0%)	
Partially performed	6 (1.4%)	2 (6.1%)	
Amnioinfusion performed	260 (58.8%)	14 (42.4%)	0.07
Amnioinfusion volume (cc)	400 (250, 725)	500 (200, 700)	0.79
Amnioreduction performed	404 (91.4%)	31 (93.9%)	1.00 *
Amnioreduction volume (cc)	1300 (775, 2220)	1725 (1225, 2125)	0.07

* Fisher’s exact test used due to small expected values in the contingency table. Data expressed as: *n* (%), median [1st quartile, 3rd quartile]. Abbreviations: LLQ—left lower quadrant, LUQ—left upper quadrant, RLQ—right lower quadrant, RUQ—right upper quadrant.

**Table 3 jcm-10-03693-t003:** Pregnancy outcomes following fetoscopic laser photocoagulation.

Pregnancy Outcome	No Septostomy (*n* = 442)	Septostomy (*n* = 33)	*p*-Value
Overall Survival			
Two Twins	334 (75.6%)	25 (75.8%)	0.95 *
One Twin	63 (14.3%)	5 (15.2%)	
No Twins	45 (10.2%)	3 (9.1%)	
CAS	36 (8.1%)	6 (18.2%)	0.06 *
PPROM	181 (41.0%)	18 (54.5%)	0.13
Iatrogenic Monoamnionicity	2 (0.5%)	7 (21.2%)	<0.01
Gestational Age at PPROM (wks)	27.3 (22.5, 31.3)	25.7 (24.7, 28.0)	0.53
PPROM before 28 wks	94 (21.3%)	13 (39.4%)	0.02
Interval from FLP to PPROM (wks)	7.0 (1.9, 10.6)	4.6 (0.4, 7.0)	0.14
IAI	30 (6.8%)	1 (3.0%)	0.71 *
Abruption	49 (11.1%)	3 (9.1%)	1.00 *
Gestational Age at Delivery (wks)	32.0 (28.9, 34.3)	29.7 (25.9, 31.9)	<0.01
Delivery before 32 wks	300 (67.9%)	30 (90.9%)	<0.01
Delivery before 24 wks	43 (9.7%)	2 (6.1%)	0.49
Interval from FLP to Delivery (wks)	11.1 (7.4, 14.1)	8.1 (4.1, 10.1)	<0.01
Interval from PPROM to Delivery (wks)	0.3 (0.0, 2.7)	1.1 (0.0, 3.7)	0.27
Indication for Delivery			
Delivery after 34 Weeks **	123 (28.8%)	1 (3.0%)	<0.01 *
Preterm Labor	146 (34.2%)	16 (48.5%)	
Maternal Indication	71 (16.6%)	9 (27.3%)	
Fetal Indication	54 (12.6%)	4 (12.1%)	
TOP/Dual Demise	11 (2.6%)	2 (6.1%)	
Other/Unknown	22 (5.2%)	1 (3.0%)	
Mode of Delivery			
Vaginal	92 (21.4%)	8 (25.8%)	0.48 *
CS	330 (76.7%)	23 (74.2%)	
Combined	8 (1.9%)	0 (0.0%)	
Single Demise	61 (13.8%)	4 (12.1%)	1.00 *
Dual Demise	5 (1.1%)	2 (6.1%)	0.08 *
Donor Demise	51 (11.5%)	6 (18.2%)	0.26 *
Recipient Demise	20 (4.5%)	2 (6.1%)	0.66 *
Donor Selective Reduction	1 (0.2%)	0 (0.0%)	1.00 *
Recipient Selective Reduction	0 (0%)	0 (0%)	1.00 *

* Fisher’s exact test used due to small expected values in the contingency table. ** Indications for delivery were only considered for deliveries less than 34 weeks. Abbreviations: CAS—chorion amnion separation, CS—Cesarean section, FLP—fetoscopic laser photocoagulation, IAI—intra-amniotic infection, PPROM—preterm premature rupture of membranes, TOP—termination of pregnancy.

**Table 4 jcm-10-03693-t004:** Neonatal outcomes.

Neonatal Outcome	No Septostomy	Septostomy	*p*-Value
Composite Morbidity **			
Two twins	30 (8.1%)	7 (25.1%)	0.02 *
One twin	63 (17.0%)	5 (17.2%)	
No twins	277 (74.9%)	17 (58.6%)	
Donor (*n* = 384)			
Neonatal death	37 (10.3%)	2 (7.7%)	1.00 *
TTN	76 (23.7%)	5 (20.8%)	0.75
RDS	118 (36.8%)	13 (54.2%)	0.09
BPD	32 (10.0%)	8 (33.3%)	<0.01 *
IVH	18 (5.6%)	1 (4.2%)	1.00 *
Sepsis	13 (4.0%)	1 (4.2%)	1.00 *
NEC	6 (1.9%)	1 (4.2%)	0.40 *
Recipient (*n* = 419)			
Neonatal death	29 (7.5%)	1 (3.3%)	0.71 *
TTN	86 (23.9%)	9 (31.0%)	0.39
RDS	139 (38.6%)	13 (44.8%)	0.51
BPD	26 (7.2%)	5 (17.2%)	0.07 *
IVH	22 (6.1%)	4 (13.8%)	0.12 *
Sepsis	16 (4.4%)	1 (3.4%)	1.00 *
NEC	10 (2.8%)	2 (6.9%)	0.22 *

* Fisher’s exact test used due to small expected values in the contingency table. Abbreviations: BPD—bronchopulmonary dysplasia, IVH—Grade III or IV intraventricular hemorrhage, NEC—necrotizing enterocolitis, RDS—respiratory distress syndrome, TTN—transient tachypnea of the newborn. ** Composite morbidity among pregnancies with one or more survivors (*n* = 399). Composite includes BPD, IVH, Sepsis, and NEC.

## Data Availability

The data presented in this study are available on request from the corresponding author. The data are not publicly available to maintain patient privacy and confidentiality.

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
