# Peer review of "Risk Factors and Outcomes Following Septostomy during Fetoscopic Surgery for Twin-to-Twin Transfusion Syndrome"

_jcm, 2021, doi:10.3390/jcm10163693_

Round 1

Reviewer 1 Report

The manuscript is well presented. The authors discuss an important topic which despite its' clinical relevance, is mostly of Interest only to the small community of FMF specialists actually performing FLP on monochorionic twins. The different centers clearly have their own preferred techniques when doing FLP, which also influences the risks of IOS - as evident for the prodded literature. This is also apparent from the presented literature and can be confirmed by our own experience in this field - supported by many personal communications between the involved centers and their specialists.

The presented work gives a good review and evaluation of the different risk factors identified for IOS. This information can and should be incorporated in counseling these families, however, in search of solution of TTTS per case, the individual anatomy and course of TTTS determines greatly which options the management group have in trying to treat TTTS (mostly determining point of entry and used method).

As a small point of recommendation, I would have been interested to learn of two further risk factors for IOS not discussed in this work:
1- the experience of the surgeon performing FLP (how the learning curve affects the risk of IOS, or how individual training and supervision is implemented)
2- additional surgical techniques (like "canule technique") routinely applied in our centre - however this may have been included in one of the tables.

I was interested to learn, that amnionfusion is so commonly performed in the presented population. Our centre is much more restrictive in this field as we consider amnioinfusion as an additional risk for all manners of membrane complications including IOS and CMS. This certainly does not diminish the merits of this manuscript, especially as this is a retrospective analysis, and was never intended as a prospective study for the method-related analysis for risks of IOS.

As the authors state, IOS seems more common in later gestational ages and with deeper vertical recent pockets, which would suggest, that iatrogenic tensioning on the intertwin membranes seems to have a negative affect on the risk of IOS.

As summary I find the manuscript well written, the data well presented and therefore recommend it for publication in present form.

Author Response

REVIEWER 1

We greatly appreciated this reviewer’s thoughtful reading and commentary on our manuscript. Our answers to reviewer thoughts and questions are in green font.

The manuscript is well presented. The authors discuss an important topic which despite its' clinical relevance, is mostly of Interest only to the small community of FMF specialists actually performing FLP on monochorionic twins. The different centers clearly have their own preferred techniques when doing FLP, which also influences the risks of IOS - as evident for the prodded literature. This is also apparent from the presented literature and can be confirmed by our own experience in this field - supported by many personal communications between the involved centers and their specialists.

The presented work gives a good review and evaluation of the different risk factors identified for IOS. This information can and should be incorporated in counseling these families, however, in search of solution of TTTS per case, the individual anatomy and course of TTTS determines greatly which options the management group have in trying to treat TTTS (mostly determining point of entry and used method).

As a small point of recommendation, I would have been interested to learn of two further risk factors for IOS not discussed in this work:
1- the experience of the surgeon performing FLP (how the learning curve affects the risk of IOS, or how individual training and supervision is implemented)
2- additional surgical techniques (like "canule technique") routinely applied in our centre - however this may have been included in one of the tables.

We see a septostomy rate between 6.6% and 7.9% (p = 0.98) for our surgeons. It is difficult to comment on how experience may make a difference because the overall rate is low and the surgeon may be reasonably experienced by the time they have enough cases to estimate their septostomy rate.

We also employ the “cannula technique” which is a way managing the situation where the angle of the trocar/scope approaches a parallel plane with an anterior placenta. The end of the trocar in gently placed, compressing anterior placenta, which “flattens it out” against the trocar end. The laser is then fired from within the trocar. Unfortunately, we have not routinely recorded use of the cannula technique at our center, and therefore cannot comment on this directly; we therefore think it is best to leave this out of the manuscript. It is worth noting that this technique is only used in cases of anterior placenta, and we do not see a difference in septostomy that relates to placental location.

I was interested to learn, that amnionfusion is so commonly performed in the presented population. Our centre is much more restrictive in this field as we consider amnioinfusion as an additional risk for all manners of membrane complications including IOS and CMS. This certainly does not diminish the merits of this manuscript, especially as this is a retrospective analysis, and was never intended as a prospective study for the method-related analysis for risks of IOS.

Historically our center has used amnioinfusion in cases where visualization is difficult for any number of reasons. In addition, some amnioinfusion volumes are quite low as fluid is injected to clear debris out of the way. These have been recorded. We have not seen a difference in outcomes (compared to other centers) based on amnioinfusion. However, we do try to avoid amnioinfusion in cases with very high starting MVPs (where it is less often needed) or when the cervix is short.

As the authors state, IOS seems more common in later gestational ages and with deeper vertical recent pockets, which would suggest, that iatrogenic tensioning on the intertwin membranes seems to have a negative affect on the risk of IOS.

As summary I find the manuscript well written, the data well presented and therefore recommend it for publication in present form.

Thank you again for the thoughtful review.

Reviewer 2 Report

This manuscript is focused on the prognosis after iatrogenic septostomy during fetoscopic laser surgery for twin-to-twin transfusion syndrome (TTTS).  
Fetoscopic laser surgery is the first-choice treatment for TTTS, but septostomy is one of the problems during the surgery, which might cause pseudomonoamnionicity. 

The authors describe clinical complications after septostomy by comparison study.

Usually, septostomy could cause during the insertion of the trocar needle. The main cause is because the trocar goes through the septum.

Therefore, it usually happens in the case of the posterior placenta. But the authors describes that placental location is the same rate in the case with septostomy.

The reviewer wonders why septostomy occured even in the anterior placental location because the trocar is inserted into the recipient's amnionic cavity.  

The authors should describe the reason why septostomy happened in the case of the anterior placenta.

Solomon technique, I guess, has become a popular method.  But, if the laser coagulation is carried via the septum to coagulate the anastomosis, which is often the case when the donor sac is collapsed, the septum might rupture due to heat degeneration.

The authors should describe the number of cases in which the trans-septum coagulation was carried out.

Author Response

REVIEWER 2

We greatly appreciated this reviewer’s thoughtful reading and commentary on our manuscript. Our answers to reviewer thoughts and questions are in green font.

This manuscript is focused on the prognosis after iatrogenic septostomy during fetoscopic laser surgery for twin-to-twin transfusion syndrome (TTTS).  
Fetoscopic laser surgery is the first-choice treatment for TTTS, but septostomy is one of the problems during the surgery, which might cause pseudomonoamnionicity. 

The authors describe clinical complications after septostomy by comparison study.

Usually, septostomy could cause during the insertion of the trocar needle. The main cause is because the trocar goes through the septum.

Thank you so much for this comment. We agree that a proportion of septostomy is caused by placement of the trocar (or the 18-gauge needle before the trocar when the Seldinger technique is used) through the collapsed donor sac. Importantly, direct inadvertent puncture of the intervening membranes or heat degeneration of the intervening membranes from the laser may also be important mechanisms. We revised lines 41 – 45 of the manuscript to make delineation of these three possible mechanisms more clear.

It is not entirely clear to us that “the main cause” of septostomy is the trocar going through the intervening membranes. We are not aware of data that make this clear as we mention in lines 236 – 238.

Therefore, it usually happens in the case of the posterior placenta. But the authors describes that placental location is the same rate in the case with septostomy.

The reviewer wonders why septostomy occured even in the anterior placental location because the trocar is inserted into the recipient's amnionic cavity.  

The authors should describe the reason why septostomy happened in the case of the anterior placenta.

Thank you for these comments. It is not clear to us that this is the case. First, other authors have reported similar rates of anterior placentation among patients with septostomy. Gordon et al. report 53% anterior placentation among patients that had iatrogenic septostomy and Peeters et al. reports 51% (References 8 and 9 from the manuscript) for example.

In our experience:

In situations with anterior placentation, the donor is often “stuck” either along the lateral or fundal uterine wall (i.e. stuck in the corner where the roof meets the wall) or along the anterior uterine wall (i.e. stuck on the roof). Often, the window for entry is much smaller when an anterior placenta is present (we try to avoid transplacental cannula/trocar placement). For the trocar placement to go through a collapsed donor sac, then donor merely needs to be near the available window. This occurs frequently regardless of placentation. An example would be a window in the left upper quadrant with the donor lying along the left lateral wall.

In addition, anterior versus posterior placentation should not affect either of the two other mechanisms by which septostomy may occur.

Solomon technique, I guess, has become a popular method.  But, if the laser coagulation is carried via the septum to coagulate the anastomosis, which is often the case when the donor sac is collapsed, the septum might rupture due to heat degeneration.

We agree, and note this as an important mechanism of iatrogenic septostomy in the manuscript (lines 41 - 46)

The authors should describe the number of cases in which the trans-septum coagulation was carried out.

We agree that this information could possibly be helpful for differentiating mechanisms of septostomy. Unfortunately, this was not a routinely collected variable. We have added this as a weakness of the manuscript (lines 265 - 267).